# What do malaria program officers want to learn? A survey of perspectives on a proposed malaria short course in Nigeria

Olufemi Ajumobi [1,2], Rotimi Felix Afolabi [3,4]*, Adefisoye Adewole[5], Muhammad Shakir Balogun[5], Patrick Nguku[5], IkeOluwapo O. Ajayi[1,3]

1 Nigeria Field Epidemiology and Laboratory Training Program, Abuja, Nigeria, 2 School of Public Health, University of Nevada, Reno, Nevada, United States of America, 3 Department of Epidemiology and Medical Statistics, College of Medicine, University of Ibadan, Ibadan, Nigeria, 4 Population and Health Research Entity, North-West University, Mmabatho, South Africa, 5 African Field Epidemiology Network Nigeria Country Office, Abuja, Nigeria

* rotimifelix@yahoo.com

**Data Availability Statement:** The data file is available from the Zenodo database (https://zenodo.org/record/5067680#.YOA2f-gzbIU).

## Abstract

### Background

In disease control, the program officers are vital to the successful implementation of control strategies. However, poor knowledge of the disease and its control, staff attrition, and lack of intentional training for new staff can lead to under-performance and ineffectiveness of interventions. Thus, the Nigeria Field Epidemiology and Laboratory Training Program, in collaboration with National Malaria Elimination Program, planned a malaria short course (MSC) to strengthen the capacity of current program managers and incoming staff. To guide the development of the curriculum for the MSC, we conducted a needs assessment survey to ascertain the perceived usefulness of the MSC, the priority rating of MSC thematic domains and associated factors.

### Methods

Overall, 384 purposively selected respondents across ten states and the Federal Capital Territory in Nigeria were interviewed. These comprised malaria and non-malaria control program staff at state, local government area (LGA) and ward levels. We administered a structured questionnaire to elicit information on socio-demographics, training needs, priority malaria thematic domains, perceived course usefulness and willingness of ministries/organizations to release staff to attend the MSC. Data were analyzed using descriptive and inferential statistics at $p<0.05$.

### Results

Mean age was 43.9 (standard deviation: 7.6 years), 172 (44.8%) were females. Of the 384 respondents, 181 (47.1%), 144 (37.5%) and 59 (15.4%) were at the ward, LGA and state levels, respectively. Seventy-two (18.8%) had never worked in malaria control program. Majority (98.7%, n = 379) reported the need for further training, 382 (99.5%) opined that the

**Funding:** This study was supported by Cooperative Agreement Number U2GGH001876 funded by the United States Centers for Disease Control and Prevention through African Field Epidemiology Network to Nigeria Field Epidemiology and Laboratory Training Program. The funders had no role in study design, data collection and analysis, decision to publish, or preparation of the manuscript.

**Competing interests:** The authors declare that they have no competing interest.

course would be useful, and all affirmed their employers' willingness towards their participation at the training. Respondents rated high the domains of basic malariology, malaria treatment, malaria prevention, surveillance/data management, use of computers, leadership skills, program management and basic statistics. Predictors of malaria topical domains' high rating were gender (odds ratio (OR) = 6.77; 95% CI:3.55–12.93) and educational qualifications (OR = 0.48; 95% CI:0.26–0.89).

## Conclusions

A malaria short course is a necessity and appropriate for program officers at different levels of health administration in Nigeria to achieve malaria elimination, taking into consideration the challenges of human resource retention. The outcome of this study should inform the curriculum and the delivery of the MSC.

## Introduction

The burden of malaria remains a concern globally with estimated 228 million cases and 405 000 deaths reported in 2018 [1]. Of these, Nigeria accounted for up to 25% of the cases and 24% of deaths [1]. The cause of persistent high burden of malaria in Nigeria is multifactorial and constitutes a challenge for the country's human resource for health [2]. These factors include high attrition and staff turnover rate, inadequate knowledge on current issues, poor public health programming skills, poor supervision [2], poor adherence to recommended treatment guideline [3, 4], lack of structured training for incoming officers and periodic re-training of staff [5]. The deployment of efficacious interventions at addressing malaria control and elimination over the years has not been accompanied with optimal uptake [6]. Data on implementation are inadequate and of poor quality and these have been attributed to poor skills of both health workers who generate the data and program managers who are the decision makers [7]. Often, staff are transferred or deployed across different programs in public health programs at the state, local government area and ward levels in Nigeria and this disrupts program activities continuity and documentation.

Program managers are vital to successful implementation of malaria control strategies [8]. A good knowledge of malaria control and elimination strategies and the capability to analyze and use data for decision making, among others, are some important attributes of a malaria program manager. While efforts have been made to develop guidelines for assessing the management and organizational capacity of national malaria control programs, gaps in malaria programming among program staff have not been adequately addressed [8]. Additionally, needs assessment data to guide the development of a malaria training course curriculum are limited.

To address these challenges, the Nigeria Field Epidemiology and Laboratory Training Program (NFELTP), in conjunction with National Malaria Elimination Program, proposed a structured malaria short course (MSC) for both malaria control program officers and officers from other disease control programs. The aim was to address the knowledge and skills gaps of staff at the programmatic level. As part of a larger effort to design the curriculum for a MSC, we conducted a mixed-methods study to assess stakeholders' opinions on the need for the course, identify priority areas to focus on and establish willingness of stakeholders' to participate [7]. The qualitative assessment established that a MSC for program managers is a priority

for malaria control effort in Nigeria and identified training needs [7]. The need assessment survey examined course content as perceived by the stakeholders. This paper describes the quantitative aspects of the perceived usefulness of the MSC and priority rating of selected thematic domains.

## Methods

### Study design and setting

This was a cross-sectional survey conducted from February to May 2018 across 10 states in Nigeria, namely: Abia, Akwa-Ibom, Bauchi, Bayelsa, Ebonyi, Ekiti, Kaduna, Kwara, Ogun and Sokoto States; and the Federal Capital Territory Abuja [FCT]. Study participants comprised malaria and non-malaria control program staff at state, LGA and ward levels to replicate the health care and political structures in each state. Participants at the ward level comprised primary health care clinical and non-clinical staff. This study was part of a larger study aimed at developing curriculum for the MSC for malaria program managers. A detailed description of the study setting has been published [7].

### Sample size determination and sampling strategy

A sample size of 384 was calculated using a conservative estimation with 50% prevalence. A two-stage sampling technique was used. At the first stage, two states in each of the six geopolitical zones were randomly selected as follows (south-west [Ekiti, Ogun], south-south [Akwa-Ibom, Bayelsa], south-east [Abia, Ebonyi], north-central [Kwara, FCT], north-west [Kaduna, Sokoto], north-east [Bauchi]. Only one state was selected in the north-east geopolitical zone because of the pervading insecurity. At stage two, participants were selected across state, LGA and ward levels using purposive sampling.

### Data collection

Trained research assistants administered a pretested structured questionnaire to collect information on socio-demographic characteristics of participants, training background, work experience, training needs, perceived usefulness of the MSC, and opinion on willingness of employer to release staff to attend the course. In addition, participants were asked to rate the malaria thematic areas/domains in order of priority for inclusion in the course. These thematic areas were basic malariology, clinical presentation of malaria, malaria diagnosis, malaria treatment, malaria prevention, surveillance/data management, use of computers, leadership skills, program management, basic statistics, communication, and ethics. Each of these domains has sub-thematic/topical areas (S1 Table). Research assistants who were NFELTP fellows (Field Epidemiologists with at least a Master of Public Health degree) administered the survey questionnaire using open data kit (ODK) software on android phones.

### Data processing and analysis

The MS Excel-generated dataset from ODK platform was analysed using IBM SPSS version 25 (Armonk, New York). Analysis commenced with data cleaning to ensure completeness and consistency. Independent variables considered were socio-demographics, course usefulness, training needs and employers' support to participate in the MSC. The main outcome variable was priority rating of malaria thematic areas/domains. The malaria thematic domains' mean scores <1.500, 1.500–2.499 and ≥ 2.500 were defined as 1-low, 2-medium, and 3- high rating, respectively. Descriptive statistics such as means and standard deviation, were used to summarise the data at univariate level and frequency distribution tables were generated. Difference in

mean scores of the rated malaria thematic domains and their respective topics were ascertained using repeated analysis of variance (ANOVA) at the bivariate level. In addition, Chi-squared and Fisher's exact tests (where applicable) were performed to assess the individual association of selected background characteristics with the malaria programming topics in each thematic domain. Ordinal logistic regression, having confirmed non-violation of proportional odds or parallel assumptions, was thereafter used to determine the influence of selected background characteristics on priority rating of malaria thematic domains at 5% level of significance.

## Ethical considerations

The study was approved by the University of Ibadan/University College Hospital Ibadan, Nigeria Institutional Review Board (Ref: UI/EC/18/0089). Individual written informed consent was obtained from each of the participants at every stage of the study. Participation was voluntary and participants could opt out of the study, as necessary. Confidentiality of responses received was maintained. There was no risk or harm to participants in this study. Data were de-identified prior to analysis.

## Results

The mean age of respondents was 43.9 (standard deviation [SD]: 7.6) years. Of the 384 respondents, 172 (44.8%) were women and 192 (50.0%) had a bachelor's degree. Nearly half of the respondents worked at ward level (47.1%, n = 181), see Table 1.

### Respondents' work experience and perception of the training course

The respondents' median year of work experience in malaria, non-malaria and any disease control program were 5 (IQR = 3–10; n = 312), 4.5 (IQR = 3–10; n = 72) and 5 (IQR = 2–9; n = 384) years, respectively. About one-third (116, 30.2%) had at least 10 years work experience (Table 2). Most respondents (n = 245; 63.8%) worked currently in malaria control program while almost one-fifth (n = 72, 18.8%) never did. Nearly all respondents reported a need

**Table 1. Frequency distribution of the background characteristics of respondents (N = 384).**

| Characteristics | Frequency |
|---|---|
|  | n (%) |
| **Age group (years)** | |
| < 35 | 48 (12.5) |
| 35–49 | 228 (59.4) |
| ≥50 | 108 (28.1) |
| **Gender** | |
| Male | 212 (55.2) |
| Female | 172 (44.8) |
| **Highest education qualification** | |
| Diploma | 139 (36.2) |
| Bachelor | 192 (50.0) |
| Postgraduate | 53 (13.8) |
| **Work Level** | |
| Ward | 181 (47.1) |
| LGA | 144 (37.5) |
| State | 59 (15.4) |

**Table 2. Respondents' work experience and perception of the training course (N = 384).**

| Characteristics | Frequency (%) |
|---|---|
| **Work experience** | |
| *Malaria control program* | |
| Never | 72 (18.8) |
| Previously | 67 (17.4) |
| Currently | 245 (63.8) |
| *Years worked in malaria and other diseases program* | |
| < 5 | 165 (43.0) |
| 5–9 | 103 (26.8) |
| ≥10 | 116 (30.2) |
| *Years worked in Malaria program (n = 312)* | |
| < 5 | 129 (41.4) |
| 5–9 | 85 (27.2) |
| ≥10 | 98 (31.4) |
| **Perception about the short course**[*] | |
| Perceived need for further training | 379 (98.7) |
| Perceived it will be useful | 382 (99.5) |
| **Employer participation** | |
| Willing to release employees to attend the course | 384 (100.0) |
| Supported the duration of the course | 358 (93.2) |

[*]multiple responses

for further training (n = 379; 98.7%) and found the proposed malaria course relevant to their respective career development/promotions (n = 382; 99.5%). Of note, all the participants confirmed their respective employers were ready to release staff to participate in the course (Table 2).

## Priority ratings of the selected topics by malaria thematic domains

The thematic domains with the highest mean score of the rated topics were leadership skill and malaria treatment while the least were ethics and clinical presentation of malaria (Table 3). The least mean score of the rated topics by the respondents were myths in recognition of malaria (2.41±0.7; in clinical presentation of malaria), conflicts of interests (2.43±0.7; in ethics) and microscopy for malaria parasite (2.43±0.7; in malaria diagnosis). In consonance with results of high rating category, topics in surveillance/data management (sources of data/data generation, 2.73±0.6; data analysis/interpretation, 2.73±0.6) and leadership skills (Mentoring, supervision, accountability, 2.73±0.6) thematic domains had the highest mean rating scores (see, S2 Table). There was no significant difference ($p>0.05$) in the ratings of topics in basic malariology, malaria treatment, program management, basic statistics, and communication thematic domains. But topics in clinical presentation in malaria, malaria diagnosis, malaria prevention, surveillance/data management, use of computer and ethics domains were rated with a significant difference ($p<0.05$) (see, Table 3).

Irrespective of the thematic domains' topics, majority (75.5%) of the participants indicated a high priority rating of all the selected malaria topics. Comparatively, the most and the least domains rated high priority were malaria treatment (85.7%) and clinical presentation of malaria (59.4%), respectively (Table 3). Overall, most participants rated the following topics high: basic malariology—continuum of malaria control (74.2%), clinical presentation of

**Table 3. Distribution of respondents' priority rating and differences in mean scores of topics by thematic domains.**

| Malaria thematic domains | Priority rating: n (%) | | | Mean score ±SD | $p$^ |
|---|---|---|---|---|---|
| | **Low** | **Medium** | **High** | | |
| Basic Malariology | 23 (6.0) | 73 (19.0) | 288 (75.0) | 2.64±0.5 | 0.0613 |
| Clinical presentation of Malaria | 24 (6.3) | 132 (34.4) | 228 (59.4) | 2.51±0.6 | <0.001* |
| Malaria Diagnosis | 28 (7.3) | 100 (26.0) | 256 (66.7) | 2.54±0.6 | <0.001* |
| Malaria Treatment | 20 (5.2) | 35 (9.1) | 329 (85.7) | 2.70±0.5 | 0.201 |
| Malaria prevention | 20 (5.2) | 96 (25.0) | 268 (69.8) | 2.53±0.5 | <0.001* |
| Surveillance/data management | 21 (5.5) | 70 (18.2) | 293 (76.3) | 2.67±0.5 | <0.001* |
| Use of computers | 20 (5.2) | 85 (22.1) | 279 (72.7) | 2.65±0.5 | 0.001* |
| Leadership skills | 23 (6.0) | 57 (14.8) | 304 (79.2) | 2.73±0.6 | na |
| Program Management | 23 (6.0) | 48 (12.5) | 313 (81.5) | 2.66±0.6 | 0.979 |
| Basic statistics | 27 (7.0) | 76 (19.8) | 281 (73.2) | 2.55±0.6 | 0.100 |
| Communication | 25 (6.5) | 93 (24.2) | 266 (69.3) | 2.59±0.5 | 0.057 |
| Ethics | 39 (10.2) | 107 (27.9) | 23 8(62.0) | 2.49±0.6 | 0.002* |
| Overall rating | *20(5.2)* | *74(19.3)* | *290(75.5)* | 2.60±0.4 | <0.001* |

*significant at p<0.05;

^ based on ANOVA test;

SD: standard deviation; na—not applicable due to the presence of only a topic

malaria—symptoms of malaria (67.2%); malaria diagnosis—reading malaria diagnostic test results (71.9%); malaria treatment—knowledge on treatment guideline (78.1%); malaria prevention—vector control (75.5%); surveillance/data management—sources of data and data generation (79.7%); use of computers—data entry and analysis (77.3%); leadership skills—mentoring, supervision, and accountability (79.2%); program management—planning activities and using resources (74.0%), and logistics and commodity distribution (74.0%); basic statistics—use of charts, graphs and tables (64.8%); communication—communication for public engagement (69.8%); and ethics—introduction to ethics (62.8%). The highest rated high priority topic was sources of data and data generation (n = 306; 79.7%), while the least was introduction to ethics (n = 241; 62.8%), see S2 Table.

## Association between respondents' characteristics and their levels of topics' priority ratings

Table 4 presents overall proportion of participants' level of topics' priority rating and its association with participants' characteristics. Proportion of participants for each level of priority rating was significantly different by gender (p<0.001), highest educational qualification (p = 0.012), work position (p = 0.003) and work status in malaria control program (p = 0.008). About 90% of the male participants reported high priority rating of topics relative to their female (58.1%) counterpart. Respondents who had current experience in malaria control program (80.8%) had high priority rating of topics compared to others who either previously (67.2%) or never (65.3%) worked in malaria control program (p = 0.008), see Table 4. However, age and years of work experience of respondents in disease control programs were not significantly associated with the topics' level of priority ratings.

**Table 4. Distribution of respondents' level of priority ratings of topics in the selected thematic domains by demographic characteristics.**

| Characteristics | Low | Medium | High | Chi-square (p-value) |
|---|---|---|---|---|
| | n (%) | n (%) | n (%) | |
| **Age group** | | | | |
| < 35 years | 4 (8.3) | 8 (16.7) | 36 (75.0) | 1.504 (0.826) |
| 35–49 years | 11 (4.8) | 43 (18.9) | 174 (76.3) | |
| ≥ 50 years | 5 (4.6) | 23 (21.3) | 80 (74.1) | |
| **Gender** | | | | |
| Male | 0 (0.0) | 22 (10.4) | 190 (89.6) | 56.54 (<0.001)* |
| Female | 20 (11.6) | 52 (30.2) | 100 (58.1) | |
| **Highest education qualification** | | | | |
| Diploma | 15 (10.8) | 24 (17.3) | 100 (71.9) | 13.870 (0.012)* |
| Bachelor | 1 (1.9) | 11 (20.8) | 41 (77.4) | |
| Postgraduate | 4 (2.1) | 39 (20.3) | 149 (77.6) | |
| **Work level** | | | | |
| Ward | 3 (1.7) | 29 (16.0) | 149 (82.3) | 16.462 (0.003)* |
| LGA | 15 (10.4) | 31 (21.5) | 98 (68.1) | |
| State | 2 (3.4) | 14 (23.7) | 43 (72.9) | |
| **Work status in malaria control program** | | | | |
| Never worked | 3 (4.2) | 22 (30.6) | 47 (65.3) | 14.113 (0.008)* |
| Previously worked | 7 (10.5) | 15 (22.4) | 45 (67.2) | |
| Currently works | 10 (4.1) | 37 (15.1) | 198 (80.8) | |
| **Years of work experience in malaria/other disease programs** | | | | |
| <5 years | 9 (5.5) | 33 (20.0) | 123 (74.6) | 1.020 (0.907) |
| 5–9 years | 5 (4.9) | 22 (21.4) | 76 (73.8) | |
| ≥10 years | 6 (5.2) | 19 (16.4) | 91 (78.5) | |

*significant at p<0.05

## Determinants of priority ratings of the selected malaria thematic domains' topics

The outcome of the ordinal logistic regression analysis conducted to identify the adjusted predictors of priority rating are shown in Table 5. Gender and educational qualification were statistically significant predictors of topical thematic domains' priority rating. After adjusting for other covariates (age, educational qualification, work level, work status and years of experience), odds of indicating a high priority rating (compared to low or medium rating) of malaria thematic domains' topics among female respondents was 6.8 times (aOR = 6.77; 95% CI: 3.55–12.93) higher than that of males. Meanwhile, participants who had a diploma certificate showed 52% lower odds (aOR = 0.48; 95% CI: 0.26–0.89) of indicating high priority rating of malaria thematic domains' topics (compared to low or medium rating) compared to those who had a postgraduate degree. Similarly, participants working at LGA level showed 48% decreased odds (aOR = 0.52; 95% CI: 0.27–0.99) of indicating a high priority rating of malaria thematic domains' topics (compared to low or medium rating) relative to those who worked at ward level. However, respondents' age group and years of work experience had no statistically significant relationship with indicating a high priority rating for topical malaria thematic domains (Table 5).

**Table 5. Adjusted ordinal logistic regression model for priority rating of selected malaria thematic domains' topics.**

| Characteristics | OR | 95% CI | p-value |
|---|---|---|---|
| **Age group (Years)** | | | |
| 35–49 | 1.74 | 0.77–3.94 | 0.185 |
| ≥50 | 2.48 | 0.96–6.39 | 0.059 |
| **Gender** | | | |
| Female | 6.77 | 3.55–12.93 | <0.001* |
| **Highest educational qualification** | | | |
| Diploma | 0.48 | 0.26–0.89 | 0.019* |
| Bachelors | 1.12 | 0.46–2.74 | 0.807 |
| **Work level** | | | |
| LGA | 0.52 | 0.27–0.99 | 0.046* |
| State | 0.64 | 0.26–1.56 | 0.327 |
| **Work status in malaria control program** | | | |
| Previously | 0.79 | 0.34–1.82 | 0.581 |
| Currently | 1.44 | 0.73–2.84 | 0.299 |
| **Years of work experience in malaria/other disease control** | | | |
| 5–9 years | 1.22 | 0.62–2.40 | 0.559 |
| ≥10 years | 1.21 | 0.62–2.36 | 0.575 |

* significant at p <0.05

## Discussion

In this study, we investigated the perception of usefulness of the MSC the priority rating of MSC thematic domains and associated factors by stakeholders in the malaria and other diseases control programs in Nigeria. It revealed nearly all the participants perceived the MSC to be useful. Specifically, the study revealed that all the selected thematic domains for the MSC had statistically significant high priority rating as the proportion of those who gave high priority rating is higher compared to either low or medium ratings. To our knowledge, this is the first quantitative study globally to examine the prioritization of thematic domains of a malaria training course and related factors from the perspectives of malaria and other diseases' stakeholders.

Most, at least 70%, rated certain topics and their domains to be of high priority, namely: parasite phases of development, malaria transmission, continuum of malaria control (basic malariology domain), knowledge on treatment guideline, treatment of malaria (malaria treatment domain), vector control, intermittent preventive treatment for pregnant women (malaria prevention domain), mentoring, supervision, and accountability (leadership skills domain), planning activities and using resources, logistics and commodity distribution, and sustainability of malaria control activities (program management domain) and basic statistics domain. The high rating of data sources and data generation, health information system, data analysis and interpretation, data utilization, use of dashboard, use of software, data entry and analysis (surveillance/data management domain) corroborate earlier finding of qualitative a study which suggested the need for data-related training and buttresses the importance of surveillance as a core intervention in the World Health Organization Global Technical Strategy for malaria [7, 9].

Confirmatory testing before treatment is the essential for rational use of antimalaria medicines [10, 11]. Reading malaria rapid diagnostic test (RDT) results was rated by most

respondents to be of high priority but not the topic on the use of RDT. This is probably because RDT the mainstay of malaria diagnosis in the country and thus respondents felt confident about awareness of it use, but not the interpretation. Over 68 million RDTs have been deployed in Nigeria [12]. Notably, use of computers domain inclusive of report writing, an essential soft skill was not rated highly by most respondents. This is probably due to the perception that service delivery should be the primary focus for malaria program.

There were significant differences in the ratings of the clinical presentation of malaria, malaria diagnosis, malaria prevention, surveillance/data management, use of computers, communication, and ethics thematic domains. Overall, in the low priority rating category, indoor residual training (IRS), conflict of interest (COI) had highest proportion of participants. This could be because IRS intervention is not commonly deployed nationwide because of its high cost and usefulness solely in the Sahel and sub-Sahel region of the country, of which only Bauchi and Sokoto states fall [6, 13]. Respondents may not have perceived the importance of COI as this tends to be more applicable to research than public health programming.

Leadership skills domain had the highest mean score, thus indicating the interest and importance of this domain to respondents. These skills are not usually taught during malaria training. Moreover, it was not surprising that leadership skills domain was rated by most respondents to be of high priority. This could be attributed to the fact that most respondents were at least 40 years of age and were expected to constitute high-level administrative officers usually in leadership positions.

The higher mean score of malaria treatment domain, signifies yearnings for knowledge, an area with dearth of knowledge over the years. Bamiselu et al. found sub-optimal adherence to treatment guidelines [3]. The finding supports that of qualitative study [7]. Generally, a high priority rating was typical of males and those who were currently working in the malaria program. Being a female and having a diploma were determinants of high priority rating of malaria thematic domains and topics. Male respondents were less likely to rate the topics high relative to female respondents, thus signifying possible influence of gender preference in the contents of the MSC. This may be because the females are more involved in the community case management of malaria and as care givers at home, they probably would like to get more information on malaria treatment.

All respondents were willing to authorize the participation of their employees and a majority supported the duration of the course. This is similar to the finding of an earlier study [7]. The respondents' perception of usefulness and readiness for the MSC revealed the course is vital for their respective career development and growth. Similar results on health workers' perceptions on malaria training and education programs have been reported recently in a qualitative study conducted in China [14]. The participants' perception of malaria training needs further buttresses the timely relevance of the proposed malaria short course. The MSC is aimed at providing evidence-based information that will help strengthen and guarantee an efficient training of health workers engaged in the Nigerian malaria program. This will eventually lead to capacity building of the malaria control program officers for program implementation, health and social research studies and training-of-trainers programs Mayor et al [15]. The first cohort of the MSC participants have been trained [7].

## Limitations

Our findings should be interpreted in the context of certain limitations. Study participants were purposively selected and thus the findings are not expected to be generalizable to other programmatic implementation settings. Moreover, the respondents exhibited similarity in the

measures of central tendency across the participants' sub-groups. The study was conducted to inform country-specific curriculum development of a MSC.

## Conclusions

A malaria short course is necessary and appropriate for program officers at different levels of health administration in Nigeria. This is to achieve malaria elimination taking into consideration the challenges of human resource retention. While we suggest the need for the implementation of a comprehensive training curriculum, the outcome of this study should be taken into consideration in planning and delivering subsequent training.

## Supporting information

**S1 Table. Malaria short course thematic domains and topics.**
(DOCX)

**S2 Table. Frequency distribution of respondents' priority rating of topics in selected malaria thematic domains.**
(DOCX)

**S1 File. Questionnaire.**
(PDF)

**S1 Dataset. Survey data.** https://zenodo.org/record/5067680#.YOA2f-gzbIU.
(XLSX)

## Acknowledgments

The authors are grateful to National Malaria Elimination Program Nigeria and the Directors of Public Health, Directors of Primary Health Care and Disease Control, Malaria Program Managers and local government area Primary Health Care Coordinators of Abia, Akwa-Ibom, Bauchi, Bayelsa, Ebonyi, Ekiti, Kaduna, Kwara, Ogun and Sokoto States and Federal Capital Territory Nigeria, for their immense support and contribution towards the success of this study. The findings of this study were presented, and feedback received at the 2018 Nigeria Centre for Disease Control /Nigeria Field Epidemiology and Laboratory Training Program Annual Scientific Conference which held in Abuja, Nigeria from 4–6 September 2018. Additionally, the authors express profound gratitude to the NFELTP fellows who assisted with data collection, namely: Oluyomi. Bamiselu, Bountain Tebeda, Ntiense. Umoette, Joseph Agboeze, Godwin Okezue, Bosede Alowooye, Istifanus Waziri, Joshua Difa, Taiwo Olasoju, Hannatu Dimas, Ismaila Ibrahim, Abubakar Danmafara, Biobelu Abaye, Tamuno-Wari Numbere, Chindima Amuzie, Nwenyi Okoro, Pius Ononigwe, Chindima Emma-Ukaegbu, Olusola Hassan Ajayi, John Ojo, Hakeem Yusuf, Olukorede Ifedolapo Ikwunne, Jibreel Omar Muhammad, Salisu Isah, Oluseyi Akano, Jenom Danjuma, Nsisong Asanga, Augustine Dada, Eric Edrah, Amina Umar, Adaora Eneja, and Irene Esu.

## Author Contributions

**Conceptualization:** Olufemi Ajumobi, IkeOluwapo O. Ajayi.

**Data curation:** Olufemi Ajumobi, Rotimi Felix Afolabi, IkeOluwapo O. Ajayi.

**Formal analysis:** Olufemi Ajumobi, Rotimi Felix Afolabi, IkeOluwapo O. Ajayi.

**Funding acquisition:** Patrick Nguku.

**Investigation:** Olufemi Ajumobi, Adefisoye Adewole, Muhammad Shakir Balogun, Patrick Nguku, IkeOluwapo O. Ajayi.

**Methodology:** Olufemi Ajumobi, Rotimi Felix Afolabi, IkeOluwapo O. Ajayi.

**Project administration:** Olufemi Ajumobi, Muhammad Shakir Balogun, Patrick Nguku, IkeOluwapo O. Ajayi.

**Resources:** Muhammad Shakir Balogun, Patrick Nguku.

**Software:** Rotimi Felix Afolabi.

**Supervision:** Olufemi Ajumobi, Muhammad Shakir Balogun, IkeOluwapo O. Ajayi.

**Validation:** Olufemi Ajumobi, Rotimi Felix Afolabi, IkeOluwapo O. Ajayi.

**Visualization:** Olufemi Ajumobi, Rotimi Felix Afolabi, IkeOluwapo O. Ajayi.

**Writing – original draft:** Olufemi Ajumobi, Rotimi Felix Afolabi, IkeOluwapo O. Ajayi.

**Writing – review & editing:** Olufemi Ajumobi, Rotimi Felix Afolabi, Adefisoye Adewole, Muhammad Shakir Balogun, Patrick Nguku, IkeOluwapo O. Ajayi.

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
