## [Decision Letter · Decision Letter 0]

28 Jun 2021

PONE-D-21-14537

What do malaria program officers want to learn? A survey of perspectives on a proposed malaria short course in Nigeria

PLOS ONE

Dear Dr. Afolabi,

Thank you for submitting your manuscript to PLOS ONE. After careful consideration, we feel that it has merit but does not fully meet PLOS ONE’s publication criteria as it currently stands. Therefore, we invite you to submit a revised version of the manuscript that addresses the points raised during the review process.

We look forward to receiving your revised manuscript.

Kind regards,

Sammy O. Sam-Wobo

Academic Editor

PLOS ONE

Journal Requirements:

3.  Please improve statistical reporting and refer to p-values as "p<.001" instead of "p=.000" or "p<0.00". Our statistical reporting guidelines are available at " ext-link-type="uri" xlink:type="simple">https://journals.plos.org/plosone/s/submission-guidelines#loc-statistical-reporting"

4. Please include additional information regarding the survey or questionnaire used in the study and ensure that you have provided sufficient details that others could replicate the analyses. For instance, if you developed a questionnaire as part of this study and it is not under a copyright more restrictive than CC-BY, please include a copy, in both the original language and English, as Supporting Information.

6. We note you have included a table to which you do not refer in the text of your manuscript. Please ensure that you refer to Table 4 in your text; if accepted, production will need this reference to link the reader to the Table.

Additional Editor Comments (if provided):

Authors to respond to comments and send back

Reviewers' comments:

Reviewer's Responses to Questions

**Comments to the Author**

1. Is the manuscript technically sound, and do the data support the conclusions?

Reviewer #1: Yes

2. Has the statistical analysis been performed appropriately and rigorously? 

Reviewer #1: Yes

3. Have the authors made all data underlying the findings in their manuscript fully available?

Reviewer #1: Yes

4. Is the manuscript presented in an intelligible fashion and written in standard English?

Reviewer #1: Yes

5. Review Comments to the Author

Reviewer #1: The manuscript is technically sound. I however suggest the following

1. that among the keywords line53 the 'program' be written as program officers

2. Use MSC in the last sentence Line 100 of the Introduction

3. line 213 correct Table 5 to Table 4

4. line 300- delete the unclear reason and replace it with- This may be because the females are more involved in home management of malaria and as care givers , they probably would like to get more information on malaria treatment.

5. line 321 Put full stop after Nigeria. This is to achieve malaria elimination, while taking into -----------

6. PLOS authors have the option to publish the peer review history of their article (what does this mean?). If published, this will include your full peer review and any attached files.

Reviewer #1: **Yes: **Chinyelu Ekwunife

---

## [Author Response · Author response to Decision Letter 0]

3 Jul 2021

The comments and observations to improving our manuscript are deeply appreciated.

Thank you!

---

## [Editor Report · Decision Letter 1]

31 Aug 2021

PONE-D-21-14537R1

What do malaria program officers want to learn? A survey of perspectives on a proposed malaria short course in Nigeria

PLOS ONE

Dear Dr. Afolabi,

Thank you for submitting your manuscript to PLOS ONE. After careful consideration, we feel that it has merit but does not fully meet PLOS ONE’s publication criteria as it currently stands. Therefore, we invite you to submit a revised version of the manuscript that addresses the points raised during the review process.

If applicable, we recommend that you deposit your laboratory protocols in protocols.io to enhance the reproducibility of your results. Protocols.io assigns your protocol its own identifier (DOI) so that it can be cited independently in the future. For instructions see: http://journals.plos.org/plosone/s/submission-guidelines#loc-laboratory-protocols. Additionally, PLOS ONE offers an option for publishing peer-reviewed Lab Protocol articles, which describe protocols hosted on protocols.io. Read more information on sharing protocols at https://plos.org/protocols?utm_medium=editorial-emailutm_source=authorlettersutm_campaign=protocols.

We look forward to receiving your revised manuscript.

Kind regards,

Sammy O. Sam-Wobo

Academic Editor

PLOS ONE

Journal Requirements:

Additional Editor Comments (if provided):

The Author is yet to respond and revise the manuscript as commented by the reviewer

Let the author respond
---

## [Author Response · Author response to Decision Letter 1]

1 Sep 2021

Indeed, we are thankful for all your efforts in ensuring we have a quality manuscript.

---

## [Editor Report · Decision Letter 2]

14 Sep 2021

What do malaria program officers want to learn? A survey of perspectives on a proposed malaria short course in Nigeria

PONE-D-21-14537R2

Dear Dr. Afolabi,

We’re pleased to inform you that your manuscript has been judged scientifically suitable for publication and will be formally accepted for publication once it meets all outstanding technical requirements.

Kind regards,

Sammy O. Sam-Wobo

Academic Editor

PLOS ONE

Additional Editor Comments (optional):

Dear Author

Minor comments have been pointed out. Attend to them and submit
---

## [Editor Report · Acceptance letter]

20 Sep 2021

PONE-D-21-14537R2 

What do malaria program officers want to learn? A survey of perspectives on a proposed malaria short course in Nigeria 

Dear Dr. Afolabi:

I'm pleased to inform you that your manuscript has been deemed suitable for publication in PLOS ONE. Congratulations! Your manuscript is now with our production department. 

Kind regards, 

on behalf of

Dr. Sammy O. Sam-Wobo 

Academic Editor

PLOS ONE